# Using mobile phone data to estimate dynamic population changes and improve the understanding of a pandemic: A case study in Andorra

**Alex Berke**[1]*, **Ronan Doorley**[1], **Luis Alonso**[1], **Vanesa Arroyo**[2], **Marc Pons**[2], **Kent Larson**[1]

**1** Media Lab, Massachusetts Institute of Technology, Cambridge, MA, United States of America, **2** Andorra Recerca + Innovació, Andorra

* aberke@mit.edu

**Data Availability Statement:** The data set and code which constitutes the minimal data set and can be used to reproduce and validate our results is made available in a public repository: https://

## Abstract

Compartmental models are often used to understand and predict the progression of an infectious disease such as COVID-19. The most basic of these models consider the total population of a region to be closed. Many incorporate human mobility into their transmission dynamics, usually based on static and aggregated data. However, mobility can change dramatically during a global pandemic as seen with COVID-19, making static data unsuitable. Recently, large mobility datasets derived from mobile devices have been used, along with COVID-19 infections data, to better understand the relationship between mobility and COVID-19. However, studies to date have relied on data that represent only a fraction of their target populations, and the data from mobile devices have been used for measuring mobility within the study region, without considering changes to the population as people enter and leave the region. This work presents a unique case study in Andorra, with comprehensive datasets that include telecoms data covering 100% of mobile subscribers in the country, and results from a serology testing program that more than 90% of the population voluntarily participated in. We use the telecoms data to both measure mobility within the country and to provide a real-time census of people entering, leaving and remaining in the country. We develop multiple SEIR (compartmental) models parameterized on these metrics and show how dynamic population metrics can improve the models. We find that total daily trips did not have predictive value in the SEIR models while country entrances did. As a secondary contribution of this work, we show how Andorra's serology testing program was likely impacted by people leaving the country. Overall, this case study suggests how using mobile phone data to measure dynamic population changes could improve studies that rely on more commonly used mobility metrics and the overall understanding of a pandemic.

## Introduction

At the start of the COVID-19 pandemic, nonpharmaceutical interventions (NPIs) were widely deployed in an effort to stymie the rate of new infections. These interventions included stay-

github.com/CityScope/CSL_Andorra_COVID_ Public. This work uses 3 main data sources: (i) public COVID-19 case reports, (ii) Andorra Telecom data, (iii) results from Andorra's serology testing program. The (i) public COVID-19 case reports data are provided within the repository. The (ii) raw Andorra Telecom data cannot be shared publicly because it is owned by Andorra Telecom and contains individual level sensitive information. Our preprocessing derives aggregated metrics from this data, which are then used in the publicly available analysis code. The preprocessing code and aggregated metrics are also available in the public repository. The (iii) serology data were collected and are managed by the Foundation for the Innovation and Research Actuatech. The dataset cannot be shared publicly because it contains individual level medical information which is sensitive. Again, our preprocessing derives aggregated metrics from this data, which are then used in the publicly available analysis code. The preprocessing code and aggregated metrics are also available in the public repository.

**Funding:** Andorra Recerca + Innovació provided financial support to the City Science Group at MIT Media Lab. The authors Alex Berke, Ronan Doorley, Luis Alonso, Kent Larson are affiliated with the City Science Group. The other authors are affiliated with Andorra Recerca + Innovació.

**Competing interests:** The authors have declared that no competing interests exist.

at-home orders and restrictions on economic activity, which were used as a means to reduce contact and hence transmission rates, effectively limiting mobility. Country border restrictions were also put in place to reduce the chance of importing the virus through inter-country travel. At the same time, tests became more available to better track population infection rates [1]. There has been an influx of data and research used to study the efficacy of various interventions [2–4]. In particular, this work addresses the use of population movement data.

Research preceding COVID-19 has indicated a close relationship exists between human mobility and the spread of infectious disease [5]. Past studies have shown how mobility data, such as commuter trips, can be used to improve disease forecasting models [6]. These earlier works highlighted the importance of combining their modeling frameworks with mobility data to address potential future emergent respiratory viruses, while also citing a lack of real-time mobility data as a limitation. In the wake of COVID-19, such real-time mobility data became widely available to study the pandemic, largely collected through airlines or via mobile phones. This is demonstrated in early works using aggregated metrics from Baidu LBS [7] to estimate domestic population movement in China. By combining this data with airline transportation data to estimate international travel, researchers modeled the effect of travel restrictions and the international spread of COVID-19 [8]. Similarly, the Baidu LBS data was also used to model the spatial spread of COVID-19 from Wuhan to evaluate the impact of domestic control measures [9]. Mobility data collected from mobile phones has also since been made available by Google [10], Facebook [11], Safegraph [12], transit apps [13], telecoms, and other companies [14]. Metrics based on these sources have been used to model or predict COVID-19 transmission rates [15–20] as well as to verify model results [21], with the assumption that changes in transmission rates are correlated with changes in the mobility metrics. Researchers have also combined mobile phone data from multiple sources to better understand the spatio-temporal dynamics of how the virus can spread. This includes work that simulated relationships between the number of virus cases imported to an area, subsequent population mobility, and virus spread in multiple European countries [22]. Whereas another study tracked a specific fast-spreading lineage of COVID-19 in the United Kingdom by combining aggregated mobility metrics from both Google and the O2 telecommunications service provider with genomic data [23].

Despite the broad use of these mobility data sources, their relationship to COVID-19 remains unclear. The published mobility metrics are often aggregated statistics representing the number of trips taken, such as measured through transit apps, or based on foot traffic to points-of-interest (POIs). Furthermore, the mobility data used in each of the above works are limited in that they report on a fraction of the population. (For example Baidu LBS and O2 have about 30% and 35% market share, respectively [9, 23], and Safegraph has one of the larger U.S. datasets yet in 2019 they covered only about 10% of the U.S. population and acknowledged reporting bias [24]). Likewise, other studies using reported cases microdata or air travel data to analyze the risks of importing the virus via inter-country travel (e.g. [25, 26]) are also limited by data sources that only report on a fraction of the true data.

## Contribution

This work presents a unique case study in Andorra, with comprehensive datasets that include telecoms data covering 100% of mobile subscribers in the country, and results from a serology testing program that more than 90% of the population voluntarily participated in. Previous work used these data sources to compare various mobility metrics and infection rates with retrospective correlation analysis [27]. This work builds upon these previous findings and develops compartmental epidemic models.

At the start of the pandemic in Andorra, border restrictions and economic lockdowns drastically reduced country entrances and internal country mobility. This study includes that period as well as when restrictions were lifted. The mobile phone data are used to estimate mobility metrics representing trips, similar to related works, as well as to conduct a real-time census and estimate metrics that represent the dynamic population changes, such as daily country entrances. These data are then used to improve the understanding of the pandemic in Andorra in multiple ways.

First, we show how Andorra's serology testing program, conducted in May 2020, was likely impacted by people leaving the country. We then show how the estimated country entrances data can improve epidemiological (SEIR) models that otherwise rely on mobility measured by trips. Related works have used meta-population SEIR models where the modeled sub-populations are dynamic, yet based on static census commuting data or based on a combination of POI visits and static commuting data (e.g. [28]). In contrast, this work uses comprehensive telecoms data to estimate a real-time census to more accurately capture the changing dynamics of the population during the period of study.

We develop and test multiple (SEIR) models that differ in how they parameterize transmission rates based on the trips and entrances metrics developed in this work. The models are simple, where their purpose is to illustrate how different types of mobility information can be better incorporated into SEIR models.

Finally, we use the best model to simulate a hypothetical counterfactual, representing a scenario where economic and border restrictions had not been put in place, and trips and entrances metrics had not drastically reduced.

## Outline

Before presenting our methods and results, we provide background information, with a timeline of events around the start of COVID-19 in Andorra, and the features of the country that contribute to a unique case study. We also provide background information about compartmental epidemic models to guide the reader in the presentation of our models.

## Background

### Andorra and COVID-19

The study region of this work is the small country of Andorra, which is located in the Pyrenees mountains and shares borders with only France and Spain. The country has a population of approximately 77,000 [29], yet attracts more than 8 million visitors annually, mostly for tourism associated with skiing and nature-related activities [30]. In addition, a large number of cross-border temporary workers reside in the country, mainly employed in the tourism industry. Andorra lacks an airport or train service so the primary way to enter or exit the country is by crossing the French or Spanish border by car. The country is divided into 7 municipalities, called parishes.

Partly because of the country's small size and limited border crossings, Andorra was able to implement comprehensive policies at the start of the COVID-19 pandemic, as well as implement a serology testing program which more than 90% of the population participated in. Furthermore, there is one telecoms provider for the entire country, which contributes a comprehensive view of all mobile subscribers who spend any time in Andorra, whether they are Andorran nationals or have foreign SIM cards. The telecoms data and serology data are used in this work and are described in the Data sources and preprocessing section.

**Timeline of COVID-19 cases and policies.**    The first COVID-19 case in Andorra was reportedly imported via Italy and confirmed March 2, 2020 [31]. Reported cases then rose rapidly in March before falling again in April (see Fig 1). On March 13, government officials

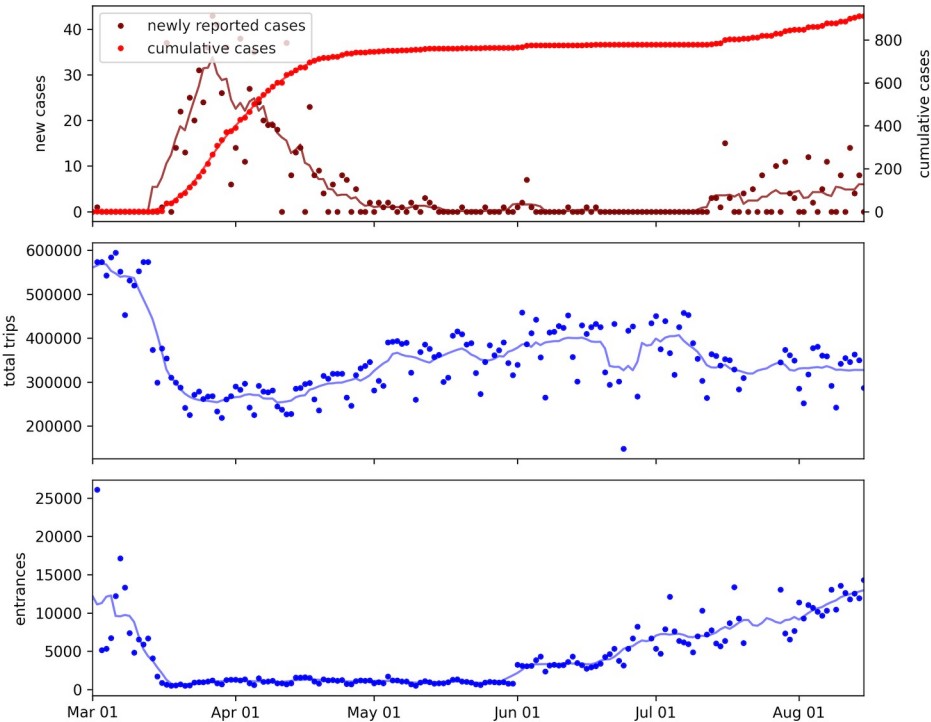

**Fig 1. Daily reported cases, trips, and entrances metrics at the start of the COVID-19 pandemic in Andorra.** The time series data are plotted for March to August, 2020, which covers the study period. Solid lines show values smoothed over a 7-day rolling window.

ordered the closure of public establishments and a quarantine was requested of the entire population. A series of COVID-19 related policies followed and neighboring country borders were restricted. In accordance with these policies, mobility within the country dropped and border crossings ceased. Other NPIs, such as masks and hand sanitizer, were also deployed. The lockdown measures in Andorra were gradually lifted in April and May, and fully lifted starting June 1. Borders also reopened in June and border crossings resumed. Table A.1 in S1 Appendix shows a timeline of COVID-19 related events.

**Nationwide serology testing program.** In May of 2020, Andorra conducted a nationwide serology testing program. This resulted in the first published seroprevalence study universally testing the entire population of a country and one of the largest of its kind [32]. Anyone over the age of 2 was invited to participate in the study, including the country's temporary workers. The testing was conducted in two phases: May 4 -14, and May 18—28, 2020. The objectives of the second phase were (a) to track the progression of COVID-19 between the two surveys and (b) to account for indeterminate or potential false negative results from the first survey. More than 90% of the population participated voluntarily in at least one of the two surveys. However, an issue with the testing program was that many participants in the first phase did not participate in the second, limiting the data collection and impact of the two-phase study. This issue is further explored and addressed in the Results section.

## SEIR models and COVID-19

SEIR models, and their variations, are compartmental models used in epidemiology. They have been widely used in forecasting COVID-19 transmission and modeling the outcomes of

government policies [15, 33, 34]. The basic concept of these models is that the population is partitioned into sequential compartments, and transitions through the compartments over time. This framework was first developed by Kermack and McKendrick in 1927 [35] and has been well described more recently by Keeling et al. [36]. In short, the SEIR model takes its name from its compartments:

$S$ = Susceptible

$E$ = Exposed

$I$ = Infectious

$R$ = Removed (quarantined, recovered, or deceased)

S represents the number of Susceptible people in the population who have not yet been exposed to the virus. Individuals transition from Susceptible to Exposed after exposure to individuals in the Infectious (I) compartment. Hence the transition S to E is a function of the number of people in the Susceptible (S) and Infectious (I) compartments, as well as the transmission rate, $\beta$, and the total population size, N. The standard model considers N constant, and the following conservation holds for any time, t:

$$N = S(t) + E(t) + I(t) + R(t) \tag{1}$$

Transitions between compartments are modeled by a set of ordinary differential equations (ODEs).

$$S'(t) = -\beta \frac{S(t)I(t)}{N(t)}$$

$$E'(t) = \beta \frac{S(t)I(t)}{N(t)} - \sigma E(t) \tag{2}$$

$$I'(t) = \sigma E(t) - \gamma I(t)$$

$$R'(t) = \gamma I(t)$$

Where

$\beta$ = transmission rate of the infection

$\sigma$ = latent rate

$\gamma$ = removal rate

The latent rate, $\sigma$, is the average rate to become infectious after exposure (i.e. $\sigma^{-1}$ = average incubation period) and the removal rate, $\gamma$, is the average rate at which individuals transition from I to R.

The modeled compartments and transitions are simplifications, yet this simple framework may be well applied to COVID-19 at the start of the pandemic, before populations were vaccinated or encountering re-infections. (Models for diseases over longer periods of time may also incorporate changes in the population via birth and death rates, while other models handle individuals becoming susceptible again [36]).

An epidemic is often characterised by the basic reproduction number, $R_0$. The estimation and value of the reproduction number is complex and often misrepresented, but in general it represents the expected number of secondary infections which would be caused by a typical infected case if everyone in the population were susceptible [37, 38]. $R_0$ can be calculated as the ratio of the transmission rate to the removal rate. Often in compartmental models, both of these parameters are constant in time. However, if one or both of these parameters is time-varying, then the variation of $R_0$ over time can be estimated. While the $R_0$ only represents the true reproductive rate at the start of the pandemic when the whole population is susceptible,

the variation of this ratio over time isolates the impact of changes in human behavior and NPIs on the reproductive rate. (The effective reproductive rate $R_t$, on the other hand, represents the actual reproductive number at any point in time, given the behaviour as well as the susceptible portion of the population [39].) Estimates for reproduction numbers have been used to understand the state of a pandemic and to measure the effectiveness of interventions [4, 34, 40–42].

$$R_0 = \beta/\gamma$$

$R_0$ is a function of both transmission rate and removal rate. The removal rate represents the rate at which infectious individuals are removed from the population and then are no longer at risk of infecting susceptible individuals. Removal might occur because they isolate, or recover and are no longer infectious, or die. The removal rate may vary due to changes in testing procedures (e.g. more proactive testing can identify more cases and cause individuals to isolate earlier in their infectious period) or government policies (e.g. quarantine rules). Likewise, the transmission rate can change due to governmental policies and behavioral changes (e.g. staying home, wearing masks, and other NPIs).

Recent models that address COVID-19 have taken into account that transmission rates vary over time [15, 43–45]. Many models do so by incorporating mobility metrics to estimate behavioral changes and model changes in transmissibility based on these data. However, these mobility metrics are often based on sources that report on a small fraction of the population, and where the mobility metrics are aggregated statistics based on the number of trips to points of interest (POIs), which may not be the most important indicators of COVID-19 transmission. This is in contrast to the telecoms data used in this work, which covers all mobile subscribers within the country of Andorra, and is provided as a complete and unaggregated dataset, not limited to trips to POIs.

We note that any of the models referenced or presented in this work are oversimplifications of the complex dynamics of disease spread. They also suffer from unreliable case reports data, limited by the availability of tests, and reactive to changes in testing protocols [1].

## Materials and methods

This section describes the SEIR models used in this work, and how they are trained and tested. It then describes data sources and preprocessing methods.

### Code and data availability

All aggregated metrics and code used in this work are made available and documented in a public repository. The code includes analysis notebooks as well as the preprocessing scripts that produced the aggregated metrics. The data reporting on individuals, which was used to compute aggregate metrics, is sensitive and kept private. https://github.com/CityScope/CSL_Andorra_COVID_Public.

### Modeling

This work develops and compares multiple SEIR models that differ in how they incorporate trips and entrances data in order to model transmission rates. The trips data measure mobility behavior within the country while the entrances data measure new country entrances (described in the Data sources and preprocessing section).

The aim is to evaluate the relative impact of the trips and entrances data on model performance; the aim is not to build a state-of-the-art, accurate predictive model. To this end, the models are highly simplified.

**Comparison models.** In SEIR models, $\beta(t)$ typically represents the average number of people an infected person would expose per-unit time if everyone were susceptible. In particular, $\beta(t)$ is used to model the transition from the Susceptible to Exposed compartments. The use of $\beta(t)$ in our models is captured by the following equation from Eq (2).

$$E'(t) = \beta(t) \frac{S(t)I(t)}{N(t)} - \sigma E(t)$$

We develop multiple models that only differ in how they define $\beta(t)$.

In the following descriptions, $b0, \ldots, bn$ are parameters of $\beta(t)$ and are estimated during model training for each model in which they are included.

One model uses trips without entrances data (model ii). Another model uses both trips and entrances data (model iii). A model that uses neither data source is used as a baseline (model i).

Each of the models use the same framework, methods, and training and testing periods, described further below.

**Model i: constant transmissibility**

This is a baseline, dummy model where $\beta(t)$ is constant.

$$\beta(t) = b0$$

**Model ii: transmission as a function of trips data**

$$\beta(t) = b0 + b1 \times trips(t)^{b2}$$

**Model iii: transmission as a function of trips and entrances data**

In this model, the average rate at which the susceptible population is exposed can be impacted by the behavior of people within the country (e.g. mobility measured in trips) as well as the import of new cases (entrances).

$$E'(t) = \beta(t) \frac{S(t)I(t)}{N(t)} + S(t)f(entrances(t)) - \sigma E(t)$$

where

$$\beta(t) = b0 + b1 \times trips(t)^{b2}$$

$$f(entrances(t)) = \frac{I(t)}{N} \times b3 \times entrances(t)^{b4}$$

$f(entrances(t))$ represents the likelihood of new country entrants importing the virus. The term $\frac{I(t)}{N}$ reflects the assumption that the likelihood of new country entrants being infectious tracks with the timeline of infection rates in Andorra. This assumption is based on the fact that during the study period, the timeline of infections in Andorra was highly correlated with the timeline of infections in Spain and France (with Pearson correlation coefficients of 0.922 (p = 0.000) and 0.932 (p = 0.000), respectively), and the primary way to enter Andorra is through the Spanish or French borders. Furthermore, telecoms data showed that 86% of entrances by foreign SIMs were either Spanish or French, and when accounting for entrances

**Fig 2. Schematic representing the SEIR model framework used in this work.** The population is divided into compartments where individuals transition through the compartments: Susceptible, Exposed, Infected, Removed, Case reported, where the transitions are described by ODEs (Eq (3)).

by Andorran SIMs, 68% of all entrances were by Spanish or French SIMs. See section A.3 in S1 Appendix.

The above functions using entrances and trips can be combined into one equivalent expression representing transmissibility. We do this to simplify modeling and maintain a common expression for $E'(t)$.

$$E'(t) = \beta(t)\frac{S(t)I(t)}{N(t)} - \sigma E(t)$$

where

$$\beta(t) = b0 + b1 \times trips(t)^{b2} + b3 \times entrances(t)^{b4}$$

**Model framework.** The SEIR framework used in this work is illustrated in Fig 2 and is described by the ODEs in Eq (3)). We note that many traditional SEIR models use the $I$ compartment to represent the entirety of an individual's infectious period. Our modeling framework assumes that individuals transition from $I$ to $R$ as soon as they suspect they are infectious. Individuals may then seek a test, and the result of the test will be reported with some delay. $C$ represents the report of a positive test after that delay, $d$.

$$S'(t) = -\beta(t)\frac{S(t)I(t)}{N(t)}$$

$$E'(t) = \beta(t)\frac{S(t)I(t)}{N(t)} - \sigma E(t)$$

$$I'(t) = \sigma E(t) - \gamma I(t)$$

$$R'(t) = \gamma I(t)$$

$$C'(t) = rR'(t - d)$$

(3)

Where

$$S(t) = N - E(t) - I(t) - R(t)$$

$C(t)$ is cumulative case reports and accounts for reporting delay, $d$, and the reporting rate, $r$.

Given initial values for the compartments and the other model parameters, time series data for the compartments can be deterministically estimated by integrating over the ODEs into the future, where each compartment time series represents the compartment population on each day, t. This is done to calibrate parameters during model training as well as to generate forecasts beyond the training period.

Initial values for $R$ and $C$ at $t = 0$ are set based on the number of cumulative reported cases at the start of the study period. Initial values for E, I, are estimated by model training, along with $\gamma$ and parameters of $\beta(t)$. The reporting rate, $r$, is set to $\frac{1}{11}$, estimated from the serology and case reports data (Data sources and preprocessing section). The latent rate, $\sigma$, is set to $\frac{1}{5.2}$, estimated by prior work [46]. The reporting delay, $d$, is set to 7, consistent with related works [21, 47] and empirical checks (see section A.7 in S1 Appendix). $d$ is the average time from when an infectious individual is removed (isolated) to the time the case is reported, and must account for the time it takes to seek a test, for the test to be processed, and for the result to be included in reported cases data. At the start of the pandemic, tests in Andorra were sent to Spain for processing, which may have increased reporting delays. The reporting delay is incorporated into the models by shifting the trips and entrances metrics time series by $d$.

See Table A.5 in section A.6 of S1 Appendix for a concise description of model parameters.

**Training and testing.** Cumulative reported cases in Andorra reached a threshold of 2 (over a 7-day average) on March 14. The serology tests, which were used to estimate the reporting rate, were conducted in May. In September, massive testing programs began and even before then, testing started to become more available. These programs and test availability increased the case identification rate, impacting both the reporting rate and the removal rate, changing the dynamics in modeling. For these reasons, the study period includes March to August, 2020. The period of March 14—May 31 is used for model training and the following 10 weeks are used for testing.

*Training.* Parameters and initial values for $E(t)$, $I(t)$ at $t = 0$ were fit with maximum likelihood estimation (MLE). Log-likelihood was computed by comparing time series values of predicted cumulative reported cases ($C$) to the time series of actual cumulative reported cases:

$$log\text{-}likelihood = \sum log\ P_{k,\lambda}(k, \lambda) \tag{4}$$

Where the sum is over all days in the training data, $P_{k,\lambda}(k, \lambda)$ is the Poisson distributed probability mass function, $k$ is actual reported cases, $\lambda$ is predicted reported cases.

Parameters were optimized by minimizing the negative log-likelihood using the L-BFGS-B method [48]. See section A.5 in S1 Appendix for details.

*Testing.* Median absolute percentage error (MAPE) over cumulative estimates has been used in a recent framework to evaluate and compare COVID-19 models [49], where the errors incorporate an intercept shift. MAPE is similarly used to evaluate and compare the performance of models in this work. Given model training estimates $S$, $E$, $I$, $R$, $C$ up to time $t$, the trained model is tested starting at time $t + 1$ as follows. The value of $C(t)$ is corrected to the true reported cases at time $t$ and further integration over the ODEs is used to continue the simulation over the test period. The resulting C estimated over the test period is compared to actual reported cases via MAPE.

## Data sources and preprocessing

Three main data sources are used in this work and are further described below: (i) serology data from the nationwide testing program conducted in May 2020, (ii) telecoms data covering all mobile subscribers in the country, (iii) official COVID-19 case and death reports. All time series metrics estimated from (ii) and (iii) are smoothed by taking the mean over a 7-day rolling window.

**Serology data.** As described in the Andorra and COVID-19 section, a nationwide serology testing program was conducted in May of 2020. The program was voluntary, and conducted in 2 phases, and 91% of the population participated.

The program was conducted for a previous research study, in which the methods and results are detailed [32]. The study was approved by the Institutional Review Board of the Servei Andorra Atencio Sanitaria (register number 0720). An anonymized version of the dataset was also provided to researchers in our lab as part of a research partnership. The dataset includes a unique identifier for each participant and results from the 1st and 2nd round of tests; test results were left empty when there was a lack of participation. The dataset also includes demographic information for participants, including their home parish and whether they are a temporary worker. As previously described, an issue with the serology testing program was that many of the participants from the first phase of testing did not participate in the second phase (see Table A.3 in S1 Appendix).

From the serology data, Bayes Theorem [50] was used to estimate the portion of the population infected up to May. With this number and the official reported cases data, we estimated a case reporting rate of $\frac{1}{11}$. This reporting rate is used in the epidemiology models described in this work.

**Telecoms data and metrics.** Andorra has one telecoms provider (Andorra Telecom), which provided the data for this study. Since they are the sole provider, the dataset covers 100% of mobile subscribers in the country, including subscribers using foreign SIM cards. This is unlike most telecoms datasets where the market is fragmented. Each data point includes a unique ID for the subscriber, a timestamp, the coordinates of the device, and nationality for the subscriber's home network. The data have been further described in [51].

The stay-point extraction algorithm of Li et al. (2008) [52] was used to reduce the series of data points for each subscriber into a series of stay-points of 10 minutes or more within a radius of 200m or less. The stay-points represent a more concise and reliable series of places the subscriber spent time; stay-points were used to infer presence in the country, dynamic population changes, and compute the trips and entrances metrics.

There are gaps in the available telecoms data and the resulting trips and entrances metrics during the period of study (data gaps are June 28–29, and July 21–27, 2020). Missing values were imputed by taking the mean across the values from the 7 days surrounding each missing period of data.

*Dynamic population inference and metrics.* On each day, a subscriber was considered present in the country if they had a stay-point in the country within a 7-day window. The window accounts for unobserved subscriber devices due to a combination of inactivity, lack of reception in certain areas, or noisy data. The beginnings and endings of periods of presence were counted as entrances to and departures from the country, respectively.

*Trips metrics.* Daily trips for subscribers were counted as their daily number of stay points minus 1, since a new stay point is recorded when a subscriber moves beyond a 200m radius. Daily trips by subscribers were summed as a total daily trips metric.

*Home inference.* The home parish of each subscriber was inferred from the telecoms data, to come up with a population count for each of the 7 parishes of Andorra. This was done by first assigning each stay-point to the parish in which it was contained. Each subscriber's home parish was then determined to be the parish in which they spent the most cumulative time during night-time hours (12:00am to 6:00am). Related studies of human mobility that use cellular data have employed similar methods [53–56].

These inferred parish-level populations were compared to the published 2020 population statistics [29]. There is a Pearson correlation coefficient of 0.959 (p < 0.001), suggesting that the telecoms data are representative of the true population. (See Table A.2 and Fig A.1 in S1 Appendix). Inferring the parish of residence is done both to check methodology as well as compare populations to serology test participation (see the Serology tests and country departures section).

## COVID-19 infection data

This dataset was made available by Johns Hopkins University [57] and downloaded from OWID [58] as a time series of daily reports. Reported cases in Andorra were used for model estimation and prediction. There were cases identified in Andorra through the May serology testing program that were reported late, on June 2 [59]. This reporting error was handled by removing the excess case reports. Fig 1 plots the resulting daily new and cumulative case reports over the period of study. Reported deaths data for Andorra and its neighboring countries, Spain and France, were used in model assumptions (see section A.3 in S1 Appendix).

## Results

### 2019 versus 2020 metrics

Before presenting our main findings, we first present the start of the pandemic in Andorra through a series of plots, and compare this period to the same period in 2019, when Andorra experienced a normal economy with tourism.

Fig 3 shows that by the start of March of 2020, there were already fewer people (mobile subscribers) in the country than in 2019. This number then substantially dropped with the start of

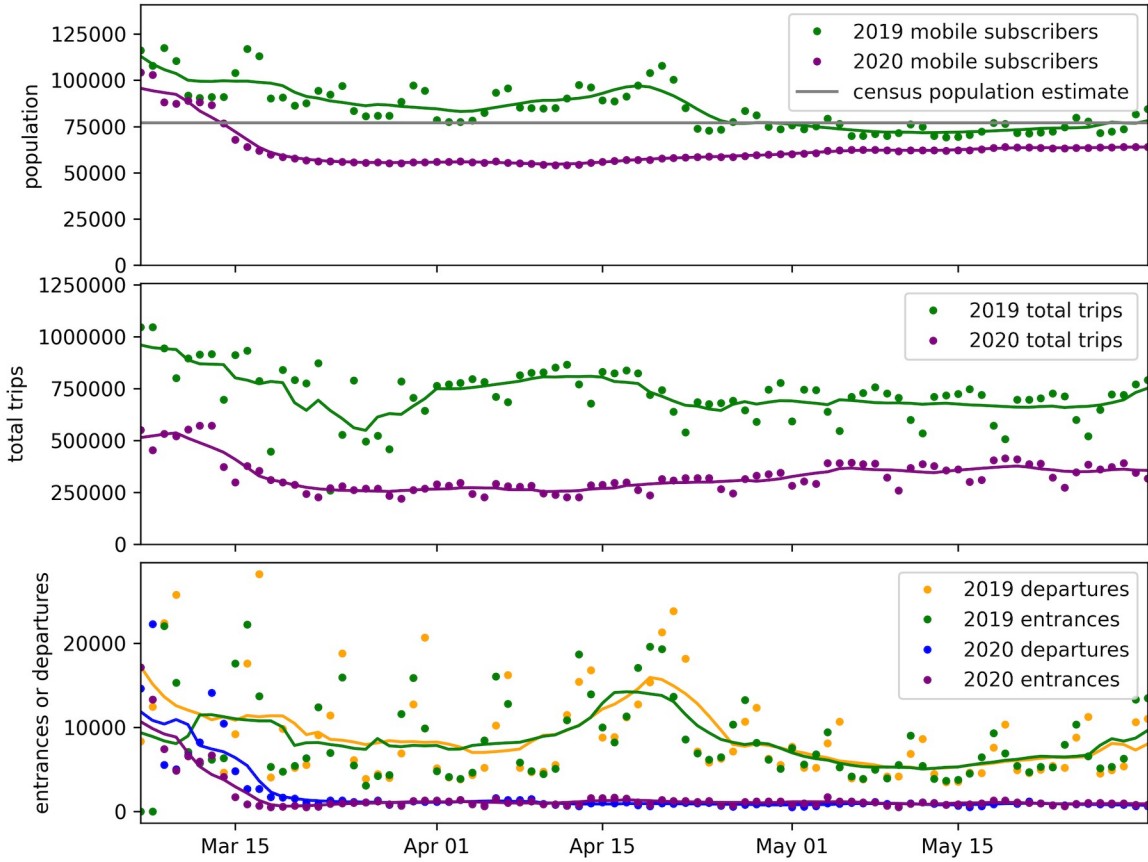

**Fig 3. Estimated population, trips, country entrances and departures metrics for 2020 vs 2019.** (Top) daily mobile subscribers counted as present in the country, (middle) daily total trips, and (bottom) daily country entrances and departures, for the country of Andorra during the start of the pandemic in 2020 versus the same period in 2019. All metrics are estimated from telecoms data that covers 100% of mobile subscribers in the country. Solid lines show values smoothed over a 7-day rolling window.

the border restrictions and economic lockdown in mid March. There were also already fewer total daily trips being taken at the start of March, 2020, compared to 2019. This is largely due to fewer people in the country making the trips. This metric also substantially dropped at the start of the lockdown. This drop was partly due to even fewer people in the country making trips, and due to the government imposing restrictions on movement. The number of trips gradually rose again before the border restrictions were lifted in June, indicating that the population increased internal mobility. The number of daily entrances to (and departures from) Andorra also significantly dropped in mid March of 2020, as tourists and others left the country and border restrictions were imposed, limiting entry to the country. These daily metrics remained near zero throughout April and May, until border restrictions were lifted in June.

## COVID-19 cases and mobility

The time series of reported COVID-19 cases is shown with the time series of the trips and entrances metrics in Fig 1. Other studies have implied that changes in case growth often lag changes in behavior and mobility metrics by 14 or more days [17, 21, 27]. However, Fig 1 shows that daily trips were able to increase throughout May of 2020 while newly reported cases remained low. Case growth did not increase again until daily entrances increased again when the border restrictions were lifted in June. This suggests that the entrances metric is more related to case growth than the trips metric in this case study. The relative predictive power of these metrics is further shown by the model results (Models results section).

## Serology tests and country departures

Andorra's nationwide serology testing program conducted in May, 2020 involved two phases of testing (see the Andorra and COVID-19 section). An issue with this program was that many of the participants from the first phase of testing did not participate in the second phase, limiting the impact of the study. An important question for a country conducting such a program might be why this happened.

This drop in participation might be particularly concerning, as we found the drop in participation was more than 3 times higher among temporary workers versus the general population, and results from the testing program showed that temporary workers had higher seroprevalence (infection rates) versus the general population. See Table A.4 in S1 Appendix. This might imply that a more infected demographic group was then less monitored.

By combining the serology test data with information inferred from the telecoms data, we find that test participants likely left the country after their first test.

We counted the number of mobile subscribers, by inferred home parish, who were in the country during the first and second phases of testing (May 4–14 and May 18–28, 2020). Subscribers were counted as present during a testing period if they had at least one "stay" within the period. We estimated how many subscribers left the country after the first test by counting how many subscribers were present during only the first test period versus both test periods.

These numbers were compared to the parish-level serology test participant populations. Namely, the portion of serology test participants who did test 1 but not test 2 was compared to the estimated portion of mobile subscribers who left the country between test periods, and this comparison was done for each home parish. Comparing across parishes, there is a statistically significant Pearson correlation coefficient of 0.937 (p = 0.0019).

To check the robustness of this result, we also restricted the May 2020 telecoms data to subscribers who had at least 7 days, or 4 nights, of data. The results are similar with Pearson correlation coefficients of 0.925 (p = 0.0028), and 0.955 (p = 0.0008), respectively.

To further validate that the decline in test participation was related to people leaving the country, we repeated these tests using 2019 telecoms data: we estimated the number of subscribers by home parish who were in the country during the periods May 4–14 and May 18–28 of 2019 (using 2019 telecoms data) and compared the number of subscribers who left the country between those periods to the serology test participation. In this case, there is a Pearson correlation coefficient of 0.4928 (p = 0.2612). If the May 2020 subscribers had left the country for reasons not related to the pandemic, we would expect the correlation to be similar for the 2019 and 2020 data. However, the correlation for the 2019 data is much lower and not statistically significant. See Table A.4 in S1 Appendix.

## Models results

Simple models based on the SEIR framework, were developed to compare the impact of trips and entrances data on transmission rates and predicted infections.

The baseline, dummy model (i) assumes a constant transmission rate. For model (ii) transmission is a function of mobility measured by trips data, and for model (iii) transmission is a function of both trips and entrances data. (See the Modeling section for details).

Models were trained over the period March 14—May 31, 2020. Table A.5 and Fig A.5 in S1 Appendix show the parameter values for the best fit models and the corresponding time series values for the estimated $R_0$, the compartment populations, and the predicted reported cases, over the training period.

Models were evaluated by their prediction performance over the weeks that followed the training period. This was done using MAPE, based on the framework used by Friedman et al. to evaluate leading COVID-19 models [49]. Results for 1—10 forecasting weeks are shown in Table 1. All models performed relatively well during the period of study. (As a point of comparison, Friedman et al. found in their global evaluation of COVID-19 models, MAPE values of 1—2% for 1 week forecasts and 17—25% for 10 week forecasts. See Figs 3 and 5 in [49].

**Table 1. MAPE results for the 3 models.**

| | MAPE | | |
| | model | | |
| forecasting weeks | i. constant $\beta$ | ii. trips data | iii. trips & entrances data |
|---|---|---|---|
| 1 | 0.03 | 0.02 | 0.24 |
| 2 | 0.20 | 0.30 | 0.19 |
| 3 | 0.65 | 0.80 | 0.25 |
| 4 | 0.97 | 1.16 | 0.34 |
| 5 | 1.22 | 1.44 | 0.52 |
| 6 | 1.47 | 1.73 | 0.76 |
| 7 | 1.36 | 1.60 | 1.01 |
| 8 | 1.39 | 1.54 | 1.08 |
| 9 | 1.59 | 1.78 | 1.09 |
| 10 | 1.80 | 1.98 | 1.12 |

Median absolute percentage error (MAPE) used to evaluate the 3 models. The MAPE measures errors relative to the true values and can vary from 0 to infinity where 0 represents perfect agreement. The models differ in whether they incorporate trips and entrances data to model transmissibility. Model (i) is a baseline, dummy model where transmissibility is constant, model (ii) uses trips data, and model (iii) uses trips and entrances data. All models used the same framework and methods.

Note their evaluation used cumulative deaths data whereas this work uses cumulative cases data.)

The model (iii) using both trips and entrances data outperformed the other models in all but excluding the first week that followed the training period. More importantly, the model (ii) that used trips data to model transmission rates (without entrances data) had results similar to, and slightly worse than, the baseline model (i) which assumed a constant transmission rate. This is not surprising, as the data indicated trips were able to increase without impacting transmission rates (Fig 1).

This is also shown in that the best fit for model (ii) had parameters that flattened the impact of the trips data, resulting in a nearly flat reproduction number, $R_0$. Given that there were few new infections at the end of the training period (i.e. a smaller population in the I compartment), this resulted in relatively flat predictions for new reported cases for model (ii) over the forecasting weeks that followed the training period (similar to model (i)). This is in contrast to the model (iii) that used both trips and entrances data, and where predictions for new reported cases closely tracked with actual predictions. See Fig 4. Overall, these estimated $R_0$ values are reasonable and within the range of values estimated by previous works [60].

As a robustness check, all models were trained and tested over an additional set of training and testing periods that ended slightly earlier than those used for the main results. (The training period for the robustness check was March 14—May 14, 2020.) The results are similar to the main results, and shown in Table A.6 and Fig A.6 in section A.8 of S1 Appendix. However in this case, the model (iii) using trips and entrances data consistently outperformed the other models for all forecasting weeks.

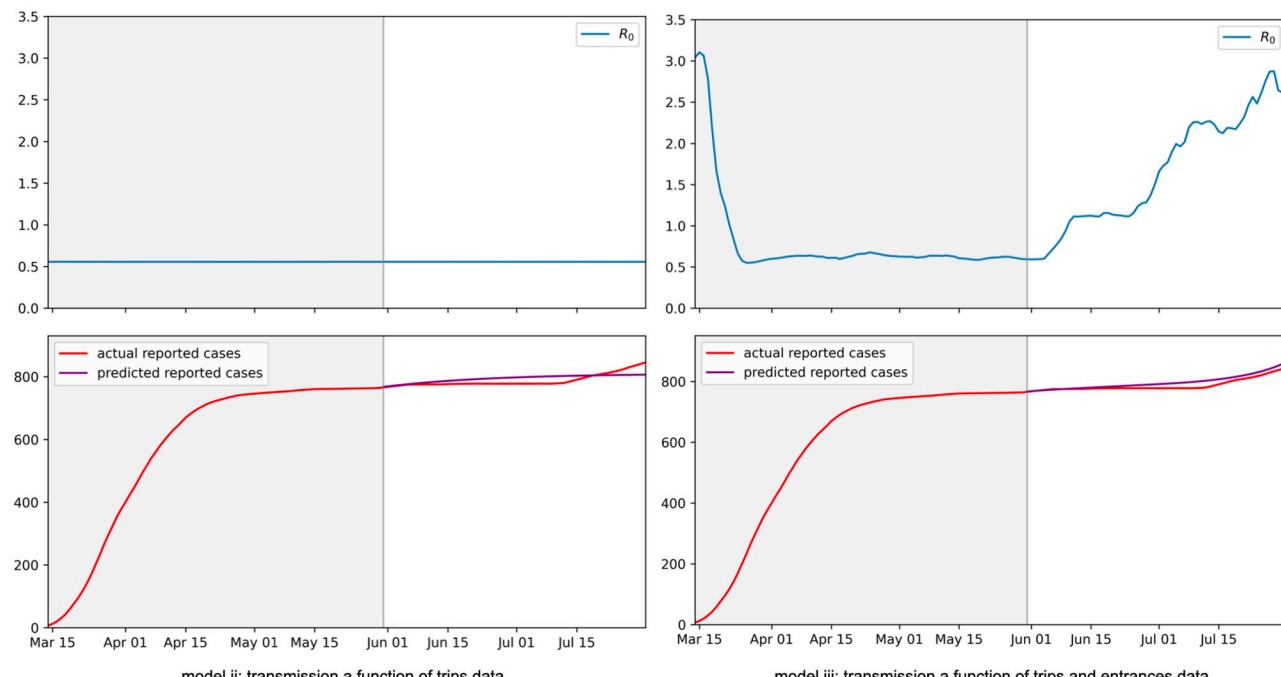

**Fig 4. Fit model results.** Time series values for (top) the estimated $R_0$ and (bottom) actual versus predicted reported cases that resulted from model training. Left: Plotted values for the model which uses just trips data. Right: Plotted values for the model which uses both trips and entrances data. Models were trained over the period March 14—May 31 and tested over the weeks that followed. The training and testing periods are divided by gray and white backgrounds, respectively. Axes for the $R_0$ values are set to highlight that values were flattened for the trips data model. See Fig A.5 in S1 Appendix for plots that show the full variation in the $R_0$ values.

These results may seem surprising and their interpretation remains unclear. In epidemiology, the 3 models may be considered as (i) a homogeneous mixing model, (ii) a model of one population in which transmission depends on local mixing only, and (iii) a model that accounts for local mixing and external seeding, where trips are a proxy for local mixing and entrances are a proxy for external seeding. It is possible that the lack of predictive power of trips in the model is due to the model being calibrated during a lockdown period, when transmission opportunities represented by trips were not as important without external seeding. However, it is also possible that while trips have been used as a proxy for mixing in related works, trips did not necessarily convert to transmission opportunities in this case. This may be due to trips being safely taken with social distancing guidelines and other NPIs in place. And again, this may partly be due to the model being calibrated during a lockdown. At the same time, the entrances metric may represent more than external seeding, and also represent a more open economy and additional activities that may increase transmission opportunities.

## Counterfactuals

What if Andorra had not imposed a lockdown, which caused reduced mobility? What if border restrictions had not been put in place, which caused a drop in entrances? Overall, what if the population mobility, measured in total trips and entrances, had not dropped in March?

In this section we explore such a counterfactual scenario by using the best fit model (iii) from the Models results section, which uses the trips and entrances data.

The lockdown in Andorra began on March 13, 2020, and there was a large drop in trips and entrances surrounding this date (see Fig 1). We again take a simplified approach to modeling, and create hypothetical trips and entrances data for a counterfactual scenario where mobility and border restrictions were not put in place. We do this by using the true metrics up to March 13 of 2020, and then keeping the metrics constant at the March 13 values. This is shown in Fig 5. We then estimate counterfactual case reports by using the previously fit model (i.e. we use the model parameters that were fit with the true trips and entrances time series values) and replace the model's trips and entrances data with the counterfactual data. We then run the simulation over the same period that was used to train the original model. The result is a prediction of 2941 cumulative reported cases up to May 31, 2020 under the counterfactual model, versus the actual 766 reported cases up to May 31, under the true scenario. The difference is an additional 2175 (more than 3x as many) reported cases during this time period under the counterfactual scenario.

## Discussion

When COVID-19 was introduced to Andorra at the start of March 2020, the country and its bordering neighbors responded quickly with economic and border restrictions. These interventions and other NPIs showed to be effective in Andorra, as the country brought case growth under control from March—May 2020, before the restrictions were fully lifted. The counterfactual scenario modeled in this work shows a stark alternative had the mobility changes observed during this period not occurred, with more than an estimated 3x as many cases, likely overwhelming the hospital system.

Numerous other works have also used mobility data collected from mobile phones to model the impacts of mobility restrictions on COVID-19 transmission. However, these studies have relied on data about trips, and the data represented a small sample. Other works using meta-population SEIR models, where the modeled sub-populations are dynamic, have been based on static census data. In contrast, this work leverages data collected from mobile phones that represent 100% of subscribers in a country.

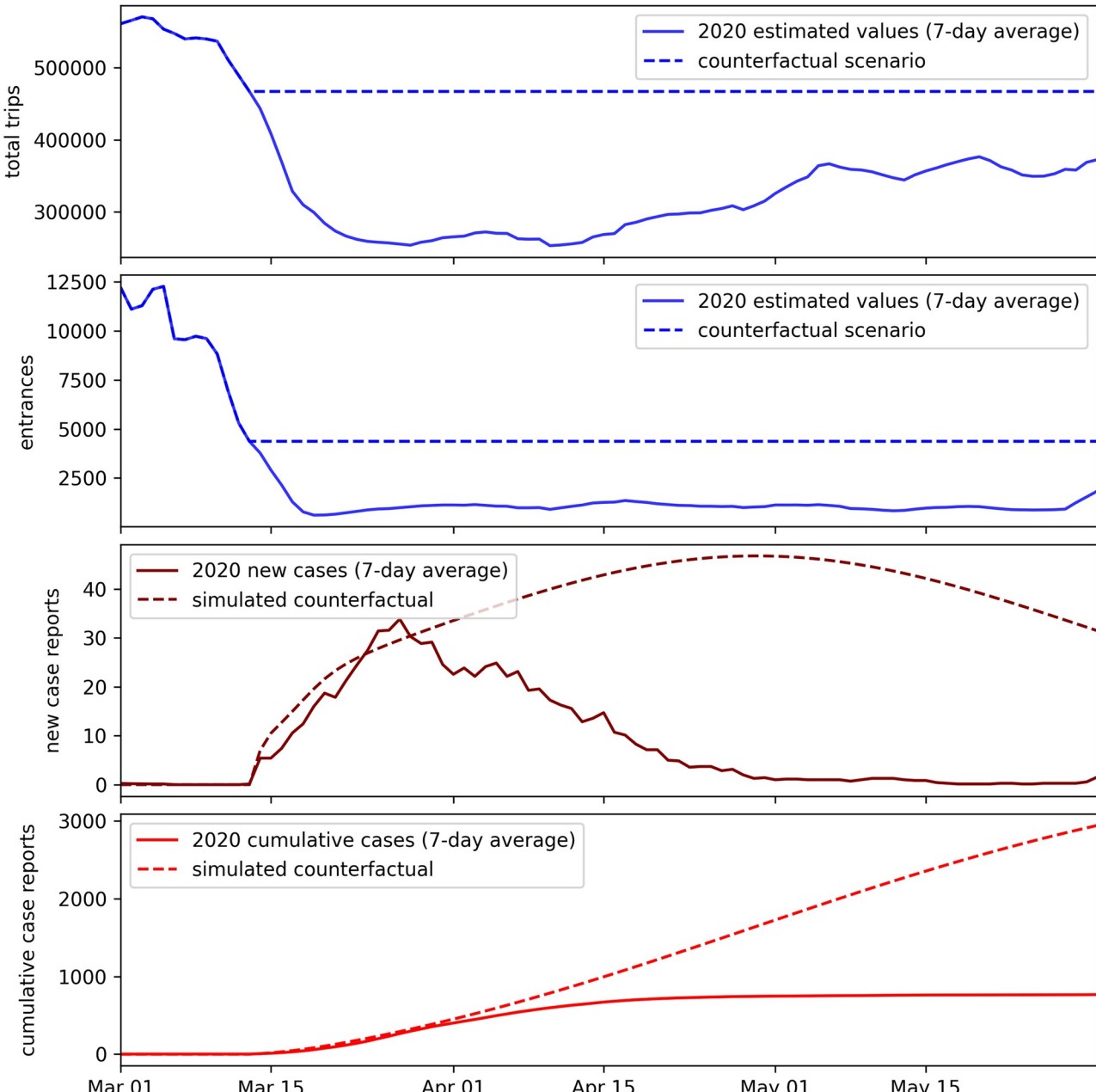

**Fig 5. Counterfactual results.** Top: Hypothetical total trips and entrances metrics that are used to simulate reported cases for a counterfactual scenario where mobility and border restrictions had not been put in place. Bottom: Simulated reported cases for such a counterfactual scenario, versus the actual reported cases that occurred in the true scenario.

We showed how these data could be used to build on previous works by computing daily trips metrics as well as estimating a dynamic, real-time population census. We then showed how these data can be used to improve upon the understanding of a pandemic in two main ways.

First, these data were used in order to better understand why participation in the nation-wide serology testing program dropped between the first and second phases of testing. The

drop in participation may have been concerning as the second phase of testing was intended to help better detect and track the virus. This decreased ability to track the virus might have been particularly concerning because the test results showed that the temporary worker population had the highest infection rates and this population also had the largest drop in test participation. However, the analysis, which leveraged the telecoms data to estimate dynamic population changes, suggested that the decline in participation was likely due to test participants leaving the country after their first test.

Second, we showed how the dynamic population data could be used to improve epidemiological (SEIR) models that otherwise rely on mobility measured by trips. In our contribution, we developed simple SEIR models that differed in how they used the trips and entrances metrics developed through this work. These models performed well compared to the 7 global COVID-19 models evaluated by Friedman et al. (2021) [15, 43, 44, 49, 61–63], but their purpose was not to be highly accurate; the purpose of these models was to illustrate the relative importance of trips mobility data versus real-time population data, namely country entrances. In particular, for the case of Andorra, we find that the population was able to regain internal mobility measured in daily total trips with limited growth in cases, and that total trips per day did not have predictive value in the SEIR models while country entrances did.

While we show that the entrances metric had superior predictive power over the trips metric in Andorra, we do not mean to draw a direct line between country entrances and new COVID-19 cases. Changes in the entrances metric may have been highly correlated with other changes that impacted transmission rates, such as changes in COVID-19 policies and cautions.

In general, the models were limited by their simplifications. For example, there was likely an interaction effect between the trips and entrances metrics that was not captured in the models. The models also assumed that the case identification rate (and hence removal rate) and reporting rate were constant, which related works have as well (e.g. [21]). However, these rates likely changed with Andorra's increased testing. Future works can more accurately model the impacts of mobility and entrances, and the interaction between these metrics. This might also include incorporating data on the infection rates for other countries whose populations contribute to entrances. Future work can also incorporate data on testing rates to better model changes in the removal and reporting rates.

Furthermore, our modeling approach was able to leverage features that make Andorra a special case study compared to other countries. In particular, Andorra normally has a highly dynamic population, given its small population and relatively large number of cross-border traffic and temporary workers. These features, along with the fact that our study was conducted over one period at the start of COVID-19, may make our results less transferable to other countries or contexts.

Despite these limitations, overall, this case study suggests how using mobile phone data to measure dynamic population changes could improve studies that rely on more commonly used mobility metrics and the overall understanding of a pandemic.

## Supporting information

**S1 Appendix. Supplementary appendix.**
(PDF)

## Author Contributions

**Conceptualization:** Alex Berke.

**Data curation:** Vanesa Arroyo, Marc Pons.

**Formal analysis:** Alex Berke, Ronan Doorley.

**Funding acquisition:** Luis Alonso, Marc Pons, Kent Larson.

**Investigation:** Alex Berke.

**Methodology:** Alex Berke, Ronan Doorley.

**Project administration:** Alex Berke, Luis Alonso.

**Resources:** Luis Alonso, Vanesa Arroyo, Marc Pons, Kent Larson.

**Software:** Alex Berke, Ronan Doorley.

**Supervision:** Kent Larson.

**Validation:** Ronan Doorley, Vanesa Arroyo, Marc Pons.

**Visualization:** Alex Berke.

**Writing – original draft:** Alex Berke.

**Writing – review & editing:** Alex Berke, Ronan Doorley, Luis Alonso, Vanesa Arroyo, Marc Pons, Kent Larson.

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
