## [Decision Letter · Decision Letter 0]

13 Dec 2021

PONE-D-21-35256Using mobile phone data to estimate dynamic population changes and improve the understanding of a pandemic: A case study in AndorraPLOS ONE

Dear Dr. Berke,

Thank you for submitting your manuscript to PLOS ONE. After careful consideration, we feel that it has merit but does not fully meet PLOS ONE’s publication criteria as it currently stands. Therefore, we invite you to submit a revised version of the manuscript that addresses the points raised during the review process.

Both referees have positively evaluated the manuscript, raising only minor comments. Please, take them into account in a revised version of the manuscript, in particular considering the concerns of Referee #1.

We look forward to receiving your revised manuscript.

Kind regards,

Michele Tizzoni

Academic Editor

PLOS ONE

Journal Requirements:

2. Please provide additional details regarding participant consent. If you are reporting a retrospective study of medical records, archived samples or third party data, please ensure that you have discussed whether all data were fully anonymized before you accessed them and/or whether the IRB or ethics committee waived the requirement for informed consent. If patients provided informed written consent to have data from their medical records used in research, please include this information.

3. Please include a complete copy of PLOS’ questionnaire on inclusivity in global research in your revised manuscript. Our policy for research in this area aims to improve transparency in the reporting of research performed outside of researchers’ own country or community. The policy applies to researchers who have travelled to a different country to conduct research, research with Indigenous populations or their lands, and research on cultural artefacts. The questionnaire can also be requested at the journal’s discretion for any other submissions, even if these conditions are not met.  Please find more information on the policy and a link to download a blank copy of the questionnaire here: https://journals.plos.org/plosone/s/best-practices-in-research-reporting. Please upload a completed version of your questionnaire as Supporting Information when you resubmit your manuscript.

4. Please include your tables as part of your main manuscript and remove the individual files. Please note that supplementary tables (should remain/ be uploaded) as separate "supporting information" files.

6. Please include a caption for figure 5.

Reviewers' comments:

Reviewer's Responses to Questions

**Comments to the Author**

1. Is the manuscript technically sound, and do the data support the conclusions?

Reviewer #1: Yes

Reviewer #2: Yes

2. Has the statistical analysis been performed appropriately and rigorously? 

Reviewer #1: Yes

Reviewer #2: Yes

3. Have the authors made all data underlying the findings in their manuscript fully available?

Reviewer #1: Yes

Reviewer #2: Yes

4. Is the manuscript presented in an intelligible fashion and written in standard English?

Reviewer #1: Yes

Reviewer #2: Yes

5. Review Comments to the Author

Reviewer #1: The manuscript highlights a very important phenomenon on the role played by cases importation on local epidemics and how this can be best used in our current epidemic models to reproduce and understand the observed trends. The authors study the epidemic in Andorra which is indeed a very nice example of the role played by seeding thanks to the high ratio of external flows versus inhabitants.

I think that the manuscript is well written and clear in the research purpose that it wants to address. The methodology and the data suit well this purpose.

I think that the manuscript is suitable for acceptance after fulfilling a minor revision.

- “The value of 149 R0 can change over the course of an epidemic due to changes in human behavior and 150 NPIs; a goal of control efforts, such as those employed during the COVID-19 pandemic, 151 is to reduce R0 to bring an epidemic under control”

The authors make some confusion on the description of R0 (which does not change), probably confusing it with Rt.

Please check the definitions to avoid misconceptions and correct the paragraph.

https://royalsociety.org/-/media/policy/projects/set-c/set-covid-19-R-estimates.pdf

- line 212: “Model ii: transmission a function of trips data” → as

- line 244: It is not clear to me why C(t) should be proportional to the removals R with a delay in time. The cumulative reported cases should be reported with a certain delay after the individuals get infected, i.e. when they start being positive to tests, not when they get recovered or they isolate or die.

- line 315: “telecom data on mobility is provided by Andorra Telecom”, but then the authors say “Furthermore, telecoms data showed that 86% of entrances by foreign SIMs were either Spanish or French, and when accounting for entrances by Andorran SIMs, 68% of all entrances were by Spanish or French SIMs”.

So the mobility data includes also foreign countries SIM cards?

- on march 163h the govt of Andorra imposed a restriction to public activities and schools.

Your model is trained on this period until May 31, can this affect the calibration of your model to more general periods, like for example periods with no activity restrictions?

- Fig3 resolution is very poor and it is very difficult to understand as it is.

- Fig4 shows a flat R0 for the trips-only model. However in Fig5 total trips seem to vary a lot during the period of observation. How come this scenario does not produce any variation of Rt with respect to trips (internal mixing)?

I understand that Andorra only counts for 77,000 inhabitants and that the external flow is very high with respect to the local population relatively to other countries, however I don’t understand how a model based only on internal trips does not provide a varying reproductive number along the time of simulation.

- line 432: only to get this clear in epidemiological terms, model 1 is a homogeneous mixing, model 2 is a model of one population in which transmission depends on local mixing only, model 3 accounts for local mixing and external seeding (importation and exportation)

- line 453: “More importantly, the model (ii) that used trips data to model transmission rates (without entrances data) had results similar to, and slightly worse than, the baseline model (i) which assumed a constant transmission rate. This is not surprising, as the data indicated trips were able to increase without impacting transmission rates”

I would say that this happens also because the model is calibrated in a lockdown period, hence internal mixing does not necessarily convert to spreading opportunities because of strong social distancing measures.

- line 454: “This is also shown in that the best fit for model (ii) had parameters that flattened the impact of the trips data, resulting in a nearly flat reproduction number, R0.” indeed, the effect of the lockdown calibration, but of course using a fixed transmission parameter leads to this problem. The authors may need to address this model scenario limitation. (check gramatic of the first part of the sentence)

- Table 1: in the description of the error metric it would be better to state also the range of the values that this metric can reach. From 0 (perfect) to N (bad).

- Fig.5, very interesting, why not plotting also the new cases time-series?

- Some recent and very related work on the effect on local epidemics of population changes and seeding caused by entrances is missing from the references, see for example:

-Mazzoli, M., Valdano, E., & Colizza, V. (2021). Projecting the COVID-19 epidemic risk in France for the summer 2021. Journal of travel medicine, 28(7), taab129.

-Kraemer, M. U., Hill, V., Ruis, C., Dellicour, S., Bajaj, S., McCrone, J. T., ... & Pybus, O. G. (2021). Spatiotemporal invasion dynamics of SARS-CoV-2 lineage B. 1.1. 7 emergence. Science, 373(6557), 889-895.

-Kraemer, M. U., Yang, C. H., Gutierrez, B., Wu, C. H., Klein, B., Pigott, D. M., ... & Scarpino, S. V. (2020). The effect of human mobility and control measures on the COVID-19 epidemic in China. Science, 368(6490), 493-497.

-Mazzoli, M., Pepe, E., Mateo, D., Cattuto, C., Gauvin, L., Bajardi, P., ... & Ramasco, J. J. (2021). Interplay between mobility, multi-seeding and lockdowns shapes COVID-19 local impact. PLoS computational biology, 17(10), e1009326.

Reviewer #2: The authors study the problem of how COVID-19 transmission dynamics relate to human mobility patterns. In particular, the aim is to compare the predictive power of domestic mobility and cross-border mobility and thus infer how important dynamic population mapping is in the context of infectious disease dynamics. This is a very interesting and relevant research question, as few studies on infectious disease spread take a dynamic population into account, often because mobility datasets are limited to a single country and do not include cross-border traffic which could inform population changes. The authors study the case of Andorra, use mobility data to inform an SEIR model, and make the surprising observation that within-country trips hardly had any predictive power, while taking into account cross-border mobility improved predictions.

The manuscript is very well written. The methods and results are described in clear, unambiguous language.

The underlying data is presented in detail, including the mobility data results of the serological study. How mobility is measured and how the data is processed is described clearly, where the authors make use of data-processing practices commonly used in literature (such as the stay detection and home detection). The serological study is properly discussed, including caveats stemming from the two waves of testing.

The data used in the study is made available online. The authors have also made software code used in the study available. I have downloaded the data and code and was able to replicate several results and figures used in the manuscript.

The SEIR model is set up, trained and evaluated appropriately. The two components of mobility are incorporated in a reasonable manner in the model. The use of three model scenarios is an appropriate way to test the influence of trips- and entrance-data on the prediction. I was honestly surprised by the extreme lack of predictive power of the model scenario using only trips data (model ii) and was very skeptical at first, but I could not find a flaw in the methodology. I do think that the entrances are such a good predictor as they are probably highly correlated with a host of other interventions that affect transmission rate, but the authors properly address this in the discussion. Altogether, I think the results, conclusions and their interpretations are sound.

I would recommend the manuscript for publication as is.

I have only one minor remark which I would appreciate being addressed by the authors:

It is fair to say that Andorra is a very special case compared to other countries, due to its very small population and relatively large amount of cross-border traffic and temporary visitors. I would assume it has a much more "dynamic" population than most countries, which might mean that the results are not very transferable to other countries, where cross-border traffic and temporary visitors are of less importance. I think it would be appropriate to address this in the discussion. It would also be interesting to include comparative values for Andorra and other countries regarding how dynamic their populations are, if available (such as the fraction of cross border traffic among all traffic, the number of tourists relative to population, number of temporary workers, etc).

6. PLOS authors have the option to publish the peer review history of their article (what does this mean?). If published, this will include your full peer review and any attached files.

Reviewer #1: No

Reviewer #2: No

---

## [Author Response · Author response to Decision Letter 0]

27 Jan 2022

Please see the document titled "Response to Reviewers" that is attached in our uploaded files.

You will find here the same content without proper formatting.

Dear editor and reviewers,

We thank you for your thoughtful review and comments. The feedback has undoubtedly improved the original manuscript. 

In the following letter we respond to each of your pieces of feedback separately.

Editor and journal requirements

Response

Please see that we have made necessary changes. We have reached out to the PLOS ONE staff and confirmed that our updated manuscript meets style requirements (we corresponded with Chloe Anderson, January 18).

2. Please provide additional details regarding participant consent.

Response

Please see the changes to our Methods section and Ethics statement that clarifies that the provided data was anonymized.

3. Please include a complete copy of PLOS’ questionnaire on inclusivity in global research in your revised manuscript.

Response

Please see the uploaded file titled 'Questionnaire on inclusivity in global research' included in our submission file.

4. Please include your tables as part of your main manuscript and remove the individual files.

Response

Please note that we have updated the tables in our manuscript and confirmed with the PLOS ONE staff that our new manuscript meets journal guidelines for tables.

5. In your Data Availability statement, you have not specified where the minimal data set underlying the results described in your manuscript can be found. 

Response

We have updated our data availability statement to be more informative. We are happy to note that both reviewers confirmed data availability. Reviewer #2 even ran the code and reproduced our results.

6. Please include a caption for figure 5.

Response

The original submission had a caption for this figure directly below Table 1, so no change has been made for this in the revised manuscript.

7. Please review your reference list to ensure that it is complete and correct. 

Response

Thank you for pointing out that we should review our reference list. In general, we observed that some of our references were not easy to find by searching for them on the web. We added URL and DOI links to the references list, where appropriate, in order to address this. We have also added the references recommended by reviewer #1.

Reviewer 1 

The manuscript highlights a very important phenomenon on the role played by cases importation on local epidemics and how this can be best used in our current epidemic models to reproduce and understand the observed trends. The authors study the epidemic in Andorra which is indeed a very nice example of the role played by seeding thanks to the high ratio of external flows versus inhabitants.

I think that the manuscript is well written and clear in the research purpose that it wants to address. The methodology and the data suit well this purpose.

I think that the manuscript is suitable for acceptance after fulfilling a minor revision.

- “The value of 149 R0 can change over the course of an epidemic due to changes in human behavior and 150 NPIs; a goal of control efforts, such as those employed during the COVID-19 pandemic, 151 is to reduce R0 to bring an epidemic under control”

The authors make some confusion on the description of R0 (which does not change), probably confusing it with Rt.

Please check the definitions to avoid misconceptions and correct the paragraph.

https://royalsociety.org/-/media/policy/projects/set-c/set-covid-19-R-estimates.pdf

Response

We thank the reviewer for their positive words. We also thank the reviewer for highlighting this potential confusion in terminology. We have updated this section of the text to clarify our use of the term R0 and to distinguish this from Rt.

- line 212: “Model ii: transmission a function of trips data” → as

Response

We thank the reviewer for suggesting this grammatical change. We have made this change to both the Model ii and Model iii descriptions.

- line 244: It is not clear to me why C(t) should be proportional to the removals R with a delay in time. The cumulative reported cases should be reported with a certain delay after the individuals get infected, i.e. when they start being positive to tests, not when they get recovered or they isolate or die.

Response

We recognize that many other SEIR works have modeled the I compartment to represent the entire time an individual is infected, and then model R as "Removed" where Removed indicates they either recovered (are no longer infectious) or died. 

However, given the guidelines and policies surrounding COVID-19, it became common for individuals to isolate themselves, and hence enter the (R) Removed compartment, as soon as they suspected themselves to be infected. For this reason, our modeling framework assumes that individuals transition from I to R as soon as they suspect they are infectious. Individuals may then seek a test, and the result of the test will be reported with some delay. We model C to represent the report of a positive test after that delay, d.

We thank the reviewer for pointing out that this was unclear. We have added additional text to the Model framework section to clarify how this assumption differs from many other SEIR models. 

- line 315: “telecom data on mobility is provided by Andorra Telecom”, but then the authors say “Furthermore, telecoms data showed that 86% of entrances by foreign SIMs were either Spanish or French, and when accounting for entrances by Andorran SIMs, 68% of all entrances were by Spanish or French SIMs”.

So the mobility data includes also foreign countries SIM cards?

Response

We thank the reviewer for pointing out this should be clarified. In the original manuscript, we mentioned in the Andorra and COVID-19 section that Andorra Telecom data covers "...all mobile subscribers who spend any time in Andorra, whether they are Andorran nationals or have foreign SIM cards".

For clarity, we have also updated the Telecoms data and metrics section to say that Andorra Telecom data covers subscribers using foreign SIMs.

- on march 163h the govt of Andorra imposed a restriction to public activities and schools.

Your model is trained on this period until May 31, can this affect the calibration of your model to more general periods, like for example periods with no activity restrictions?

Response

We thank the reviewer for pointing out that we should better address this. In the Training and testing subsection we explain the data complications that cause us to use this period. The training period of March 14 - May 31 includes both severe restrictions as well as a gradual reopening of the economy. In the Model results subsection we explain that we also conduct a robustness check over a modified training and testing period, which does not include the Phase 3 reopening period, where results are summarized and then provided in further detail in the SI Appendix.

To further address the reviewer's point we have added text in the Model results section to comment on the impact of calibrating the model during the lockdown period. We have also added a comment in the discussion that notes how limiting our study to this specific period is a limitation to transferring our results to other contexts.

- Fig3 resolution is very poor and it is very difficult to understand as it is.

Response

We thank the reviewer for pointing this out. We have revised the figure and hope the reviewer finds it more informative.

- Fig4 shows a flat R0 for the trips-only model. However in Fig5 total trips seem to vary a lot during the period of observation. How come this scenario does not produce any variation of Rt with respect to trips (internal mixing)?

I understand that Andorra only counts for 77,000 inhabitants and that the external flow is very high with respect to the local population relatively to other countries, however I don’t understand how a model based only on internal trips does not provide a varying reproductive number along the time of simulation.

Response

We note that while Fig 4 shows a flattened R0, this is due to the scale of the plot. There is variation which is fully shown in Fig A.5 in S1 Appendix. We call the reader's attention to these plots in the Model results subsection and particularly in the caption of Fig 4. 

Even so, we agree with the reviewer that this flattened result with the trips data is surprising and reason as follows. During our modeling period, which followed Andorra's lockdown, we found that trips were not correlated with transmission rate (as shown in Fig 1). Due to the lack of predictive power of internal trips in the model, variation in these trips did not impact variation in the transmission rates. Since transmission rate was parameterized on trips only in this model, the modeled transmission rate was effectively almost constant. This counter-intuitive result demonstrated the need to incorporate more information - such as the entrances data - into the modeling of transmission rate.

As for how it could be the case that trips did not impact the modeled transmission rates, we have added text to address this following your comments below.

- line 432: only to get this clear in epidemiological terms, model 1 is a homogeneous mixing, model 2 is a model of one population in which transmission depends on local mixing only, model 3 accounts for local mixing and external seeding (importation and exportation)

- line 453: “More importantly, the model (ii) that used trips data to model transmission rates (without entrances data) had results similar to, and slightly worse than, the baseline model (i) which assumed a constant transmission rate. This is not surprising, as the data indicated trips were able to increase without impacting transmission rates”

I would say that this happens also because the model is calibrated in a lockdown period, hence internal mixing does not necessarily convert to spreading opportunities because of strong social distancing measures.

- line 454: “This is also shown in that the best fit for model (ii) had parameters that flattened the impact of the trips data, resulting in a nearly flat reproduction number, R0.” indeed, the effect of the lockdown calibration, but of course using a fixed transmission parameter leads to this problem. The authors may need to address this model scenario limitation. (check gramatic of the first part of the sentence)

Response

We thank the reviewer for suggesting these clarifications. We have added text to the Model results section that describes the 3 models in the epidemiological terms that the reviewer recommends. "In epidemiology, the 3 models may be considered as (i) a homogeneous mixing model, (ii) a model of one population in which transmission depends on local mixing only, and (iii) a model that accounts for local mixing and external seeding, where trips are a proxy for local mixing and entrances are a proxy for external seeding." We then add text to further interpret the results and address the fact that the lack of predictive power in the trips metric may be due to calibrating the model during a lockdown period.

- Table 1: in the description of the error metric it would be better to state also the range of the values that this metric can reach. From 0 (perfect) to N (bad).

Response

In the revised manuscript, the following explanation has been added to the caption:

“The MAPE measures errors relative to the true values and can vary from 0 to infinity where 0 represents perfect agreement."

- Fig.5, very interesting, why not plotting also the new cases time-series?

Response

We are glad that the reviewer found Figure 5 and the corresponding analysis interesting. Following the reviewer's suggestion, we have updated this figure to include a plot with the new cases timeseries.

- Some recent and very related work on the effect on local epidemics of population changes and seeding caused by entrances is missing from the references, see for example: …

Response

We thank the reviewer for calling our attention to these related works. We have incorporated each of them into the Introduction section.

Reviewer 2

The manuscript is very well written. The methods and results are described in clear, unambiguous language.

The underlying data is presented in detail, including the mobility data results of the serological study. How mobility is measured and how the data is processed is described clearly, where the authors make use of data-processing practices commonly used in literature (such as the stay detection and home detection). The serological study is properly discussed, including caveats stemming from the two waves of testing.

The data used in the study is made available online. The authors have also made software code used in the study available. I have downloaded the data and code and was able to replicate several results and figures used in the manuscript.

The SEIR model is set up, trained and evaluated appropriately. The two components of mobility are incorporated in a reasonable manner in the model. The use of three model scenarios is an appropriate way to test the influence of trips- and entrance-data on the prediction. I was honestly surprised by the extreme lack of predictive power of the model scenario using only trips data (model ii) and was very skeptical at first, but I could not find a flaw in the methodology. I do think that the entrances are such a good predictor as they are probably highly correlated with a host of other interventions that affect transmission rate, but the authors properly address this in the discussion. Altogether, I think the results, conclusions and their interpretations are sound.

Response

We thank the reviewer for both their skepticism and for thoroughly reviewing our methods and using our open source code and data to replicate results.

I would recommend the manuscript for publication as is.

I have only one minor remark which I would appreciate being addressed by the authors:

It is fair to say that Andorra is a very special case compared to other countries, due to its very small population and relatively large amount of cross-border traffic and temporary visitors. I would assume it has a much more "dynamic" population than most countries, which might mean that the results are not very transferable to other countries, where cross-border traffic and temporary visitors are of less importance. I think it would be appropriate to address this in the discussion. It would also be interesting to include comparative values for Andorra and other countries regarding how dynamic their populations are, if available (such as the fraction of cross border traffic among all traffic, the number of tourists relative to population, number of temporary workers, etc).

Response

We agree with the reviewer that our modeling approach, and hence results, are particularly well suited to a country like Andorra, where there is relatively high cross-border traffic compared to the stable population. We thank the reviewer for suggesting that we note this in the discussion and have added a paragraph describing this.

---

## [Decision Letter · Decision Letter 1]

18 Feb 2022

Using mobile phone data to estimate dynamic population changes and improve the understanding of a pandemic: A case study in Andorra

PONE-D-21-35256R1

Dear Dr. Berke,

We’re pleased to inform you that your manuscript has been judged scientifically suitable for publication and will be formally accepted for publication once it meets all outstanding technical requirements.

Kind regards,

Michele Tizzoni

Academic Editor

PLOS ONE

Additional Editor Comments (optional):

Reviewers' comments:

Reviewer's Responses to Questions

**Comments to the Author**

1. If the authors have adequately addressed your comments raised in a previous round of review and you feel that this manuscript is now acceptable for publication, you may indicate that here to bypass the “Comments to the Author” section, enter your conflict of interest statement in the “Confidential to Editor” section, and submit your "Accept" recommendation.

Reviewer #1: All comments have been addressed

Reviewer #2: All comments have been addressed

2. Is the manuscript technically sound, and do the data support the conclusions?

Reviewer #1: Yes

Reviewer #2: Yes

3. Has the statistical analysis been performed appropriately and rigorously? 

Reviewer #1: Yes

Reviewer #2: Yes

4. Have the authors made all data underlying the findings in their manuscript fully available?

Reviewer #1: Yes

Reviewer #2: Yes

5. Is the manuscript presented in an intelligible fashion and written in standard English?

Reviewer #1: Yes

Reviewer #2: Yes

6. Review Comments to the Author

Reviewer #1: All my comments have been addressed, I have no further comments and I find the manuscript suitable for publication

Reviewer #2: (No Response)

7. PLOS authors have the option to publish the peer review history of their article (what does this mean?). If published, this will include your full peer review and any attached files.

Reviewer #1: No

Reviewer #2: No

---

## [Editor Report · Acceptance letter]

24 Feb 2022

PONE-D-21-35256R1 

Using mobile phone data to estimate dynamic population changes and improve the understanding of a pandemic: A case study in Andorra 

Dear Dr. Berke:

I'm pleased to inform you that your manuscript has been deemed suitable for publication in PLOS ONE. Congratulations! Your manuscript is now with our production department. 

Kind regards, 

on behalf of

Dr. Michele Tizzoni 

Academic Editor

PLOS ONE